# Design and Psychometric Analysis of the COVID-19 Prevention, Recognition and Home-Management Self-Efficacy Scale

**DOI:** 10.3390/ijerph17134653

**Published:** 2020-06-28

**Authors:** José Manuel Hernández-Padilla, José Granero-Molina, María Dolores Ruiz-Fernández, Iria Dobarrio-Sanz, María Mar López-Rodríguez, Isabel María Fernández-Medina, Matías Correa-Casado, Cayetano Fernández-Sola

**Affiliations:** 1Nursing Science, Physiotherapy and Medicine Department, Faculty of Health Sciences, University of Almeria, 04120 Almería, Spain; j.hernandez-padilla@ual.es (J.M.H.-P.); mrf757@ual.es (M.D.R.-F.); ids135@ual.es (I.D.-S.); mlr295@ual.es (M.M.L.-R.); Isabel_medina@ual.es (I.M.F.-M.); mcc249@ual.es (M.C.-C.); cfernan@ual.es (C.F.-S.); 2Adult, Child and Midwifery Department, School of Health and Education, Middlesex University, London NW4 4BT, UK; 3Associate Researcher, Faculty of Health Sciences, Universidad Autónoma de Chile, Temuco 4780000, Chile; 4Clinical Manager, Internal Medicine Ward (COVID-19 area), Hospital de Poniente, 04700 Almería, Spain

**Keywords:** COVID-19, psychometrics, self-efficacy

## Abstract

In order to control the spread of COVID-19, people must adopt preventive behaviours that can affect their day-to-day life. People’s self-efficacy to adopt preventive behaviours to avoid COVID-19 contagion and spread should be studied. The aim of this study was to develop and psychometrically test the COVID-19 prevention, detection, and home-management self-efficacy scale (COVID-19-SES). We conducted an observational cross-sectional study. Six-hundred and seventy-eight people participated in the study. Data were collected between March and May 2020. The COVID-19-SES’ validity (content, criterion, and construct), reliability (internal consistency and test-retest reliability), and legibility were studied. The COVID-19-SES’ reliability was high (Cronbach’s alpha = 0.906; intraclass correlation coefficient = 0.754). The COVID-19-SES showed good content validity (scale’s content validity index = 0.92) and good criterion validity when the participants’ results on the COVID-19-SES were compared to their general self-efficacy (r = 0.38; *p* < 0.001). Construct validity analysis revealed that the COVID-19-SES’ three-factor structure explained 52.12% of the variance found and it was congruent with the World Health Organisation’s recommendations to prevent COVID-19 contagion and spread. Legibility analysis showed that the COVID-19-SES is easy to read and understand by laypeople. The COVID-19-SES is a psychometrically robust instrument that allows for a valid and reliable assessment of people’s self-efficacy in preventing, detecting symptoms, and home-managing COVID-19.

## 1. Introduction

The coronavirus disease (COVID-19) pandemic has caused a public health emergency worldwide [1]. COVID-19 is a potentially fatal disease caused by the severe acute respiratory syndrome coronavirus 2 (SARS-CoV-2) [1,2,3]. The clinical presentation of COVID-19 can vary widely. Whilst most patients seem to remain asymptomatic or present with mild to moderate upper respiratory tract illness [4,5], some develop severe viral pneumonia with respiratory failure that can lead to death [1,2,3,6]. According to initial reports, many people with COVID-19 require hospitalisation or critical care [3,5,7,8,9]. Consequently, many countries have implemented strict public health measures to contain the spread of COVID-19 [10]. All these public health measures rely on the general public’s ability to adopt protective behaviours to avoid the contagion and spread of COVID-19 (for example, hand hygiene and social distancing) [10,11]. Since some of these public health measures are considered disruptive to people’s day-to-day life [12], their behavioural responses should be studied. However, little is known about people’s behavioural response to adopt all the recommended protective behaviours amidst the COVID-19 pandemic.

### Background

Available evidence suggests that COVID-19 is a highly transmittable disease [13,14,15]. It is estimated that one case of COVID-19 could cause between 3 and 6 secondary cases during the outbreak [15]. In addition, early reports indicate that 7–16% of all COVID-19 cases may need hospitalisation and 5–12% could need ICU admission [3,5,7,8,9]. As evidence suggests that critical care capacities could be exceeded due to the COVID-19 pandemic [16,17], many governments worldwide have prioritised public health measures to control the spread of COVID-19 and flatten the peak of morbidity and mortality caused by the pandemic [10,18,19]. Following the World Health Organisation’s (WHO) advice [11,20], governments have urged the general public to implement protective behaviours to avoid the contagion and spread COVID-19 [10,18,19]. Some of these recommendations require individuals to be able to maintain social distancing, identify COVID-19 symptoms, follow local protocols to seek healthcare advice, or even manage people with mild symptoms of COVID-19 at home under strict preventative measures to avoid the contagion and spread [11,20]. In addition, more waves of the COVID-19 pandemic are expected in the future [21,22], which may lead to having to adopt the above-mentioned protective behaviours for long periods of time. Therefore, healthcare professionals and behavioural scientists should study the general public’s protective behavioural responses amidst the COVID-19 pandemic [19].

According to the protection motivation theory (PMT), self-efficacy is considered a robust predictor of various health-related behaviours [23,24]. Bandura’s Social Cognitive Theory defines self-efficacy as one’s perceived ability to perform a target behaviour [25,26]. Research has shown that self-efficacy could predict the general public’s intention to engage in social distancing during a simulated pandemic caused by a respiratory infectious disease [24]. Furthermore, higher self-efficacy levels have been linked to taking other preventive measures during the SARS and influenza A pandemics (for example: handwashing, respiratory hygiene, and wearing a mask when having symptoms) [27,28,29]. In this regard, an individual’s self-efficacy in preventing, recognising symptoms, and home-managing COVID-19 should be studied and analysed. Nevertheless, although numerous instruments to assess behavioural responses during an epidemic have been used [28,30,31], there are no validated tools that would allow for a valid and reliable assessment of individuals’ self-efficacy in preventing, recognising symptoms, and home-managing a respiratory infectious disease. The aim of this study is to develop and psychometrically test the COVID-19 prevention, recognition, and home-management self-efficacy scale (COVID-19-SES).

## 2. Materials and Methods

### 2.1. Design

We completed the study in two phases following an observational cross-sectional design. In the first phase, we generated the tool’s items and pilot tested its content validity and reliability (i.e., internal consistency and test-retest reliability). In the second phase, we assessed the COVID-19-SES’ validity (i.e., content, criterion and construct validity), reliability (i.e., internal consistency and test-retest reliability), and legibility.

### 2.2. Participants and Sampling

We used a convenience sampling method to recruit 678 participants. All participants had to be at least 18 years old and accept voluntary participation before taking part in the study. In line with expert guidelines [32,33,34], we recruited between 50–60 participants for the pilot study (*N* = 56) and more than 10 participants per item for the final validation study (*N* = 622). Individuals who participated in the pilot study did not take part in the final validation study.

### 2.3. Data Collection

We collected data between March and May 2020 through online surveying. We generated an online questionnaire with three different sections. In the first section, we asked the participants to provide some demographic information (i.e., age, gender, level of education completed, occupation, history of respiratory or cardiovascular chronic conditions, perceived health, and experience of COVID-19 symptoms). In the second section, we asked the participants to complete the COVID-19-SES. In the last section, we asked the participants to complete the General Self-Efficacy Scale (GSES) [35]. For the pilot study, the participants completed the same online questionnaire twice with a 2-week interval. In the final validation study, the participants were asked to complete the COVID-19-SES twice with a four-week interval. This allowed us to explore the COVID-19-SES test-retest reliability.

### 2.4. Ethical Considerations

The study was approved by the competent Ethics and Research Committee (protocol reference code: 76/2020). In the introductory section of the online survey, we provided information about the study (i.e., justification, aim, and methods), our research group, and participants’ right to withdraw at any point. We also explained how we were going to safeguard their anonymity and confidentiality. All data were treated according to the European legislation on data protection. Only three members of the research team had access to the raw data generated by the online survey platform (M.D.R.-F., I.D.-S., and M.C.-C.). They were in charge of coupling the participants’ test-retest responses and deleting the participants’ emails from the initial database. Then, the complete initial database (without email addresses) was handed to two different researchers (I.M.F.-M, M.M.L.-R.), who transferred all the data into a SPSS database and randomised the order of the responses. Once the SPSS database was created, the participants’ responses were deleted from the cloud and all the researchers’ laptops. Only the principal researcher has access to the blind SPSS database used for data analysis. All participants accepted voluntary participation before completing the online questionnaire by ticking a box that stated, “I agree to participate in the study described above and give my consent for my responses to be used with research purposes.”

### 2.5. Phase 1: Item Generation and Pilot Study of the COVID-19-SES

#### 2.5.1. Item Generation

According to the PMT, both the effectiveness of recommended behaviours and one’s self-efficacy to perform such behaviours strongly influence people’s motivation to engage in health-preventive conducts [23,24]. In line with the PMT, we generated the COVID-19-SES’ items based on the WHO’s recommended behaviours to protect oneself and others from the spread of COVID-19 [11,20]. Firstly, the research team analysed all the WHO recommendations and identified three categories in which to group the items: (1) prevention of COVID-19 spread and contagion, (2) early recognition of COVID-19 symptoms, and (3) home-management of patients with (or suspected) COVID-19 [11,20]. Secondly, we summarised the WHO recommendations and created the initial 19-item version of the COVID-19-SES. The 19 items comprising the COVID-19-SES were created by consensus. Following Bandura’s Social Cognitive Theory, people’s self-efficacy to adopt protective behaviours was measured in terms of their own capabilities to perform such behaviours [25,26]. In line with Bandura’s recommendations, the response options ranged from 0 (“completely sure that I cannot do it”) to 100 (“completely sure that I can do it”) [25,36]. In addition, we added some degree of difficulty to all the items in order to avoid a ceiling effect (for example: “in any context” at the end of item 4 and “even if my professional or social life are at stake” at the end of item 5) [37].

#### 2.5.2. Pilot Study Methods

Before conducting a final validation study, we explored the COVID-19-SES’ content validity using a panel of experts and then tested its reliability after administering it to a pilot sample (*N* = 56).

##### Content Validity

We followed Polit & Beck’s recommendations to explore the COVID-19-SES’ content validity [32]. Firstly, we recruited a panel of 14 experts from 5 different institutions in Spain and the UK. All the experts were either physicians or nurses with more than 10 years’ experience in public health, epidemiology, or infectious diseases. Secondly, we asked the experts to rate the 19 items comprising the COVID-19-SES depending on their relevance to measure people’s self-efficacy in preventing, detecting symptoms, and home-managing COVID-19 (1 = not relevant; 2 = somewhat relevant; 3 = quite relevant; 4 = highly relevant) [31]. Lastly, we calculated each item’s content validity index (i-CVI) by adding the number of experts who rated the item as “quite relevant” or “highly relevant” and dividing it by the total number of experts [32]. We considered that an i-CVI ≥ 0.78 was appropriate [32].

##### Reliability

To pilot test the COVID-19-SES, we administered the questionnaire to 56 participants and explored the tool’s internal consistency and test-retest reliability. We decided that an item positively contributed to the COVID-19-SES’ internal consistency if its corrected item-total correlation index (C-ITC) was higher than 0.3 and the tool’s Cronbach’s alpha (α) did not increase significantly after removing that item [34]. Additionally, we computed the scale’s α and considered this as appropriate if it was higher than 0.7 [32]. To explore the COVID-19-SES’ test-retest reliability, we computed the intraclass correlation coefficient (ICC) between the participants’ scores on the test and on the retest we performed two weeks later. An ICC > 0.5 was deemed as appropriate [32].

### 2.6. Phase 2: Final Validity, Reliability, and Legibility Analysis of the COVID-19-SES

Following Streiner & Kottner’s recommendations, we tested the COVID-19-SES’ validity (i.e., content, criterion and construct) and reliability (i.e., internal consistency and test-retest reliability) [34]. Additionally, we explored the scale’s legibility and developed an internal scoring system [32,38]. We analysed the data using IBM^®^ SPSS Statistics^®^ v.25 (IBM Company, Chicago, IL, USA). The COVID-19-SES was created and tested in Spanish. Four independent translators participated in the forward-backward procedure to translate the English version of the COVID-19-SES presented in this paper [39]. Two independent native English translators (proficient in Spanish) translated the COVID-19-SES from Spanish to English. The small differences between their translations were reconciled by consensus and a single English version of the COVID-19-SES was created. Two independent native Spanish translators (proficient in English) performed a backtranslation of the COVID-19-SES English version into Spanish. Again, small differences in their backtranslation were reconciled by consensus. The research team and the translators examined the original COVID-19-SES, its English translation, and the backtranslation in Spanish. It was unanimously considered that the English version of the COVID-19-SES included in this paper respected the semantics of the original tool.

#### 2.6.1. Validity

##### Content Validity

We explored the COVID-19-SES’ content validity following the same method as the one described in phase 1. In addition, we calculated the scale’s content validity index (S-CVI). We deemed appropriate a S-CVI higher than 0.78 [32].

##### Criterion Validity

In order to explore the instrument’s criterion validity, we correlated the participants’ results on the COVID-19-SES with their results on the General Self-Efficacy Scale (GSES) [35]. Prior to this, we performed a normality test on the variable “mean_score_COVID”. The kurtosis (value = 1.78; SE = 0.20), skewness (value = −1.21; SE = 0.10), Shapiro-Wilk test (W(622) = 0.91; *p* < 0.001), and histogram showed that the variable was not normally distributed [40]. Therefore, we calculated the Spearman correlation coefficient (r) between the participants’ scores on the COVID-19-SES and on the GSES [32,41,42]. A correlation coefficient (r) higher than 0.3 was considered acceptable.

##### Construct Validity

To explore the COVID-19-SES’ construct validity, we carried out an exploratory factor analysis (EFA) with principal axis factoring (PAF) and varimax rotation. Firstly, we computed the Kaiser-Mayer-Olkin test (KMO) and Bartlett’s test of sphericity to explore the adequacy of the analysis. A KMO > 0.7 and a significant Bartlett’s test (χ^2^) were considered as indicators of the appropriateness to conduct an EFA [32,41,43]. The number of factors extracted as underlying dimensions of the COVID-19-SES was determined by: (1) the point where the curve of the scree plot clearly levelled off and (2) the number of factors with eigenvalues equal or higher than 1 [32,41,43]. Items were considered to be part of a factor (or dimension) when their factor-loading coefficient was ≥ 0.45 [43].

#### 2.6.2. Reliability

We explored the COVID-19-SES’ reliability (i.e., internal consistency and test-retest reliability) following the same method as the one described in phase 1. Furthermore, we studied the correlations (Spearman’s correlation coefficient) between the COVID-19-SES and all its sub-dimensions [32].

#### 2.6.3. Legibility

We analysed the COVID-19-SES’ legibility using the INFLESZ scale, which measures the difficulty of healthcare-related texts directed at laypeople [44]. According the INFLESZ scale, texts can be very easy (> 80), quite easy (66–80), normal (56–65), somewhat difficult (40–55), or very difficult (< 40) [44]. In addition, we included a text box in the online questionnaire for participants to express whether they had any difficulties reading, understanding, or completing the COVID-19-SES.

#### 2.6.4. Scoring and Interpretation System

In order to facilitate the interpretation of the results yielded from the COVID-19-SES, we developed an internal scoring system following the experts’ recommendations [32,38]. In this regard, we calculated the participants’ mean score on the COVID-19-SES and its standard deviation (SD), which allowed us to create the following five scoring categories: (1) very low self-efficacy = scores > 2 SD below the mean, (2) low self-efficacy = scores between 1–2 SD below the mean, (3) moderate self-efficacy = scores ≤ 1 SD below the mean, (4) high self-efficacy = scores ≤ 1 SD above the mean, and (5) very high self-efficacy = scores > 1 SD above the mean.

## 3. Results

### 3.1. Pilot Study Results

Table 1 summarises the results of the pilot study. The experts considered that all the items were relevant to measure the intended construct (i-CVI > 0.78) and they were all kept as part of the COVID-19-SES that we administered to the pilot sample (*N* = 56). The COVID-19-SES’ α was 0.905 and it would not have increased if we had removed any of the items. All the items’ C-ITC was higher than 0.3. Furthermore, the average measure ICC was 0.917 with a 95% confidence interval from 0.86 to 0.91 (F(55,55) = 12.00, *p* < 0.001). All items were kept as part of the COVID-19-SES for its final validation study.

### 3.2. Participants and Descriptive Data

Table 2 summarises the participants’ characteristics. The participants’ mean age was 35.80 (SD = 13.89) and they were 68.8% female (*n* = 428). Almost 40% of the participants had completed primary or secondary education, 31% had completed vocational training, and around 30% had completed a university degree, masters or PhD. In terms of participants’ occupation, almost 25% were unemployed (*n* = 153) and 20% were healthcare professionals (*n* = 125). The majority of participants had not been diagnosed with a respiratory or cardiovascular chronic condition (*n* = 543) and had not experienced COVID-19 symptoms (*n* = 411). Most participants declared their health to be good (*n* = 352) or very good (*n* = 202). Table 3 and Table 4 show the participants’ mean score on each item and each sub-dimension of the COVID-19-SES.

### 3.3. Validity

Content validity analysis showed that all the items’ i-CVI > 0.78 (See Table 3) and the COVID-19-SES’ S-CVI was 0.92. In terms of criterion validity, our analysis indicated that the participants’ scores on the COVID-19-SES moderately correlated with their scores on the GSES (r = 0.38; *p* < 0.001). Regarding the COVID-19-SES’ construct validity, the KMO test (KMO = 0.904) and the Bartlett’s test of sphericity (χ^2^(171) = 6585.145; *p* < 0.001) evidenced the appropriateness for an EFA to be conducted. Our PAF analysis showed that the COVID-19-SES has three underlying factors that explained 52.12% of the cumulative variance found (see Table 4). These three factors presented an eigenvalue > 1 and all the items loaded onto one factor with a factor-loading coefficient > 0.45. Table 4 shows the COVID-19-SES’ dimensional structure: [Factor 1] prevention of COVID-19 contagion and spread, [Factor 2] recognition of COVID-19 symptoms, and [Factor 3] home-management of people with COVID-19 symptoms.

### 3.4. Reliability

Table 3 presents the items’ C-ITC and the scale’s α if the items were removed. The COVID-19-SES’ α was 0.906, all the items’ C-ITC > 0.3, and the scale’s α would not have increased if we had removed any of the items. Additionally, all the sub-dimensions’ α was higher than 0.70 (see Table 5). Regarding test-retest reliability, 85% of participants completed the retest after 4 weeks (*n* = 531) and we found that the average measure ICC was 0.757 with a 95% confidence interval from 0.71 to 0.80 (F(530,530) = 4.12, *p* < 0.001). Lastly, Table 6 shows that all the correlations between the participants’ mean total score for the COVID-SES-19 and its sub-dimensions were higher than 0.30.

### 3.5. Legibility

The results from the INFLESZ analysis (score = 68.04) showed that the COVID-19-SES was “quite easy” to read and understand by laypeople. Furthermore, none of the participants reported any difficulties to read, understand, or complete the scale.

### 3.6. Scoring and Interpretation System

The participants’ mean score on the COVID-19-SES was 83.32 (SD = 13.24). We developed the following internal scoring system: (1) very low self-efficacy = scores below 55, (2) low self-efficacy = scores 55–68, (3) moderate self-efficacy = scores 69–82, (4) high self-efficacy = scores 83–96, and (5) very high self-efficacy = scores above 96.

## 4. Discussion

The aim of this study was to develop and psychometrically test the COVID-19 prevention, recognition, and home-management self-efficacy scale (COVID-19-SES). Bandura’s theoretical framework suggests that higher levels of self-efficacy indicate individuals’ higher motivation towards carrying out a given task [25,26]. In fact, self-efficacy has been linked to better preventive behavioural responses in a pandemic outbreak [24,27,28,29]. In an attempt to control the pandemic, many governments have urged the general public to confine themselves at home and/or to adopt protective behaviours that can alter their day-to-day life [10,12,18,19]. Developing an instrument for the assessment of individuals’ self-efficacy to adopt preventive measures would allow healthcare professionals to explore the general public’s behavioural responses amidst the public health emergency generated by the COVID-19 pandemic [19].

When a psychometric instrument is developed, we have to carefully examine its validity and reliability [32,34]. Whilst validity refers to the instrument’s ability to actually measure the construct that it intends to measure [33,41], reliability refers to its ability to produce accurate and consistent results across time [32,33]. In this study, we assessed the COVID-19-SES’ validity and reliability. Furthermore, we analysed its legibility and created a scoring system to facilitate the interpretation of the results [32,38].

Regarding the COVID-19-SES’ validity, we analysed its content, criterion, and construct validity [34]. A panel of 14 independent experts critically reviewed the COVID-19-SES and decided that all the items comprising the tool were relevant to measure individuals’ self-efficacy in preventing, recognising symptoms and home-managing COVID-19 [41]. In terms of criterion validity, our analysis showed that the participants’ results on the COVID-19-SES moderately correlated to their results on the general self-efficacy scale. Self-efficacy is situation-specific and people can be more or less efficacious in some realms than others [25,45]. Although both instruments assess people’s self-efficacy, they both refer to very different realms in their lives and this could explain why we found a moderate correlation between them. With regard to our construct validity analysis, we found that the COVID-19-SES has a three-dimension structure. The three dimensions of the COVID-19-SES allow for the assessment of people’s self-efficacy in preventing the contagion and spread of COVID-19, recognising COVID-19 symptoms, and home-managing patients with mild COVID-19 symptoms. These three dimensions are in line with the WHO’s recommendations for the general public to avoid the contagion and spread of COVID-19 [11,20]. In general, the results from our validity analysis suggest that the COVID-19-SES is a valid instrument for assessing the general public’s self-efficacy in their ability to adopt preventive behaviours to prevent, recognise symptoms, and home-manage COVID-19 [32,33,41]. In terms of reliability, our analysis has shown that all the items of the COVID-19-SES contribute to its strong internal consistency. Additionally, our test-retest reliability analysis suggests that the COVID-19-SES can yield consistent results across time. This evidence indicates that the COVID-19-SES can measure the general public’s self-efficacy in preventing, recognising symptoms, and home-managing COVID-19 in a reliable and consistent manner [32,33,41]. Contributing to the good psychometric properties of the tested instrument, our legibility analysis showed that the COVID-19-SES is easy to read, understand, and complete by laypeople [44]. Lastly, our descriptive analysis showed that our participants are highly efficacious in preventing, recognising symptoms, and home-managing COVID-19. This could be related to the fact that the Spanish government has implemented strict measures to oblige the general public to implement protective measures against the contagion and spread of COVID-19 [19]. Future research should use the COVID-19-SES to collect data about different populations’ self-efficacy in order to deepen our understanding of its mediating effect on people’s behavioural responses to the pandemic over time.

Although we conducted a methodologically rigorous study, we need to highlight some limitations. Firstly, our sample was selected through a convenience sampling technique, which means that our results cannot be generalised. Although participants stated they were from 33 different provinces in Spain, we suggest that future studies use a stratified sampling method to recruit a representative sample. Secondly, our sample mainly comprised of young, healthy adults. It is important to test the COVID-19-SES’ main psychometric properties amongst samples of people with specific clinical, social, or behavioural backgrounds. Thirdly, the COVID-19-SES was developed and validated in Spanish. Although the instrument complies with the WHO’s recommendations to prevent the contagion and spread of COVID-19, its use in a different language should be preceded by a psychometric evaluation. Fourthly, due to the strict lockdown measures implemented by the Spanish government during the COVID-19 outbreak, we could only collect data through online surveying. Populations with limited access to the Internet or social media may be unrepresented. Lastly, data were collected through self-administered questionnaires. Since the Spanish government decreed the state of alarm and people were obliged to comply with some of the preventive behaviours reflected in the COVID-19-SES [12], the study participants could have fallen into social desirability bias.

## 5. Conclusions

Following an exhaustive assessment, the COVID-19-SES has shown to have good psychometric properties. Our results suggest that the COVID-19-SES is a valid, reliable, and legible instrument that would allow for the assessment of people’s self-efficacy in preventing, recognising symptoms, and home-managing COVID-19. Healthcare professionals and behavioural scientists should use the COVID-19-SES to study both people’s level of confidence in their ability to adopt protective behaviours amidst future waves of the pandemic as well as the relationship between self-efficacy and people’s behavioural responses in a pandemic caused by an infectious respiratory disease. The results yielded from the COVID-19-SES could provide information about people’s motivation to comply with the recommended protective behaviours. Therefore, healthcare professionals could use the COVID-19-SES when they require patients with (or suspected) mild COVID-19 symptoms and relatives to isolate themselves at home. Low levels of self-efficacy in preventing, recognising symptoms, and home-managing COVID-19 could indicate the need for health education interventions. Future studies should focus on testing the COVID-19-SES amongst different populations in different socio-cultural contexts and confirming the tool’s dimensionality.

## Figures and Tables

**Table 1 ijerph-17-04653-t001:** Internal consistency and content validity results from the pilot study (*N* = 56).

	i-CVI ^1^	NP-SES’ Alpha if Item Deleted	C-ITC ^2^
**Item 1.** Regularly and thoroughly wash my hands with soap and water or clean them with an alcohol-based hand sanitiser wherever I go.	1	0.906	0.316
**Item 2.** Cover my mouth and nose with a tissue or my bent elbow every time I cough or sneeze.	1	0.904	0.394
**Item 3.** Not touch my eyes, nose, or mouth under any circumstances.	0.93	0.903	0.475
**Item 4.** Maintain at least one metre distance between myself and others at all time.	0.93	0.903	0.459
**Item 5.** Avoid getting in contact with large groups of people even if my professional and social life are at stake.	1	0.902	0.490
**Item 6.** Only go outside when permitted and following the government’s directions.	1	0.905	0.373
**Item 7.** Identify if I have symptoms of COVID-19 quickly after they appear.	0.93	0.899	0.657
**Item 8.** Decide when symptoms require me to either call the COVID-19 emergency phone number or go to see a doctor, following the recommendations from the health authorities.	0.93	0.905	0.389
**Item 9.** Decide when my situation requires me to either call the COVID-19 emergency phone number or continue with my normal life, according to recommendations from health authorities.	0.86	0.903	0.466
**Item 10.** Call the correct phone number that the health authorities of my region have enabled for COVID-19 emergencies.	1	0.902	0.505
**Item 11.** Isolate persons with symptoms in a well-ventilated room for exclusive use, no matter how hard this may be.	0.86	0.901	0.537
**Item 12.** Ensure that the waste from the person with symptoms goes into a self-closing rubbish bin with a sealed bag, which is not shared with other household members.	0.93	0.898	0.644
**Item 13.** Reserve, if possible, a bathroom for the exclusive use of the person with symptoms.	0.79	0.907	0.461
**Item 14.** Keep the door to the room of the person with symptoms closed at all times.	0.93	0.896	0.703
**Item 15.** Limit the movement of the person with symptoms in the house, even if it is sometimes difficult.	0.93	0.892	0.828
**Item 16.** Maintain a minimum distance of 1 metre from the person with symptoms at all times.	0.86	0.894	0.753
**Item 17.** Ensure that the person with symptoms wears a mask and gloves every time they leave the room, without exception.	0.79	0.898	0.653
**Item 18.** Carry out an exhaustive daily cleaning following experts’ recommendations regarding material, disinfectant products, water temperature, and important surfaces.	1	0.894	0.798
**Item 19.** Remove the waste from the person with symptoms following experts’ safety recommendations.	0.93	0.896	0.707

^1^ i-CVI = Content Validity Index of each item; ^2^ C-ITC = Corrected Item-Total Correlation.

**Table 2 ijerph-17-04653-t002:** Participants’ sociodemographic characteristics.

Characteristics	Sample (*N* = 622)
	M ± SD
**Age** (years)	35.80 ± 13.89
	*N* (%)
**Gender**	
Female	428 (68.8)
Male	190 (30.5)
**Highest educational level completed**	
Primary	70 (11.3)
Secondary	167 (26.8)
Vocational qualification	195 (31.4)
University degree	157 (25.2)
Masters or PhD	33 (5.3)
**Occupation**	
Unemployed	153 (24.6)
Healthcare professional	125 (20.1)
Qualified worker	222 (35.7)
Non-qualified worker	111 (17.8)
Retired	11 (1.8)
**Have you experienced COVID-19 symptoms?**	
Yes	18 (2.9)
No	411 (66.1)
I am not sure	193 (31.0)
**Respiratory or cardiovascular chronic condition**	
Yes	79 (12.7)
No	543 (87.3)
**Perceived general health**	
Very poor	1 (0.2)
Poor	5 (0.8)
Normal	62 (10.0)
Good	352 (56.6)
Very good	202 (32.5)

M = Mean; SD = Standard Deviation.

**Table 3 ijerph-17-04653-t003:** Internal consistency and content validity results from the final validation study (*N* = 622).

	i-CVI	NP-SES’ Alpha if Item Deleted	C-ITC	M ± SD
**Item 1.** Regularly and thoroughly wash my […].	1	0.906	0.364	88.90 ± 17.15
**Item 2.** Cover my mouth and nose with a […].	1	0.905	0.448	92.30 ± 14.15
**Item 3.** Not touch my eyes, nose or mouth […].	0.93	0.905	0.425	68.30 ± 24.85
**Item 4.** Maintain at least one metre […].	0.93	0.906	0.397	79.70 ± 22.55
**Item 5.** Avoid getting in contact with large […].	1	0.907	0.329	90.30 ± 17.94
**Item 6.** Only go outside when permitted […].	1	0.905	0.423	94.00 ± 14.12
**Item 7.** Identify if I have symptoms […].	0.93	0.904	0.459	79.60 ± 20.99
**Item 8.** Decide when symptoms require […].	0.93	0.904	0.480	84.10 ± 20.49
**Item 9.** Decide when my situation requires […].	0.86	0.903	0.513	84.40 ± 19.66
**Item 10.** Call the correct phone number […].	1	0.904	0.477	89.20 ± 18.23
**Item 11.** Isolate the person with symptoms […].	0.86	0.896	0.744	82.20 ± 24.73
**Item 12.** Ensure that the waste from the […].	0.93	0.897	0.701	77.90 ± 28.03
**Item 13.** Reserve, if possible, a bathroom […].	0.79	0.908	0.464	72.10 ± 37.32
**Item 14.** Keep the door to the room of the […].	0.93	0.897	0.718	81.10 ± 27.63
**Item 15.** Limit the movement of the person […].	0.93	0.895	0.778	81.70 ± 25.46
**Item 16.** Maintain a minimum distance […].	0.86	0.897	0.719	79.90 ± 26.44
**Item 17.** Ensure that the person with […].	0.79	0.900	0.626	81.50 ± 27.77
**Item 18.** Carry out an exhaustive daily […].	1	0.897	0.741	82.90 ± 23.12
**Item 19.** Remove the waste from the person […].	0.93	0.896	0.743	81.40 ± 24.40

i-CVI = Content Validity Index of each item; C-ITC = Corrected Item-Total Correlation; M = Mean; SD = Standard Deviation.

**Table 4 ijerph-17-04653-t004:** Exploratory factor analysis results and structure of the COVID-19-SES (*N* = 622).

	FACTOR		
	1	2	3
Prevention of COVID-19 contagion and spread			
Item 1	0.090	0.172	**0.533**
Item 2	0.189	0.138	**0.569**
Item 3	0.185	0.164	**0.518**
Item 4	0.191	0.058	**0.531**
Item 5	0.093	0.020	**0.594**
Item 6	0.208	0.180	**0.451**
Recognition of COVID-19 symptoms			
Item 7	0.183	**0.648**	0.167
Item 8	0.133	**0.891**	0.143
Item 9	0.188	**0.842**	0.144
Item 10	0.231	**0.543**	0.208
Home-management of people with COVID-19 symptoms			
Item 11	**0.803**	0.160	0.193
Item 12	**0.704**	0.146	0.256
Item 13	**0.555**	0.076	0.057
Item 14	**0.803**	0.157	0.136
Item 15	**0.873**	0.158	0.162
Item 16	**0.757**	0.133	0.228
Item 17	**0.622**	0.171	0.222
Item 18	**0.644**	0.249	0.358
Item 19	**0.679**	0.231	0.324
% of variance	26.31	13.66	12.15
% of cumulative variance	26.31	39.97	52.12

The factor loading figures in bold indicate which factor each item loads onto.

**Table 5 ijerph-17-04653-t005:** Cronbach’s alpha and descriptive data for the COVID-19-SES and its sub-dimensions.

	Cronbach’s Alpha	M ± SD
COVID-19-SES	0.905	83.32 ± 13.24
Prevention of COVID-19 contagion and spread	0.726	85.58 ± 12.27
Recognition of COVID-19 symptoms	0.852	84.31 ± 16.54
Home-management of people with COVID-19 symptoms	0.919	80.09 ± 13.24

M = Mean; SD = Standard Deviation.

**Table 6 ijerph-17-04653-t006:** Correlations between COVID-19-SES and its sub-dimensions.

	Prevention	Recognition	Home-Management
Prevention	-	-	-
Recognition	0.389 *	-	-
Home-management	0.491 *	0.444 *	-
Total COVID-19-SES	0.721 *	0.746 *	0.849 *

* significant at the 0.01 level (2-tailed).

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
