# Peer review of "Design and Psychometric Analysis of the COVID-19 Prevention, Recognition and Home-Management Self-Efficacy Scale"

_ijerph, 2020, doi:10.3390/ijerph17134653_

Round 1

Reviewer 1 Report

This study is a paper on reliability and validity of measurement questions. The importance of the study is recognized by COVID-19. However, it is required to revise the following points.

This study uses only some of the methods that are statistically used to verify reliability and validity. Analysis using structural equation models, etc., should also be added.

Second, although the discussion of reliability and validity is discussed, he interpretation of the answer to the question itself is overlooked. It needs more description of the survey results.

Author Response

REVIEWER 1

Reviewer’s comment: This study is a paper on reliability and validity of measurement questions. The importance of the study is recognized by COVID-19. However, it is required to revise the following points. This study uses only some of the methods that are statistically used to verify reliability and validity. Analysis using structural equation models, etc., should also be added.

Authors’ response: Thanks for your comment. If we understood well, the reviewer suggests conducting a confirmatory factor analysis (CFA) using structural equation modelling (SEM). We recognise the importance of confirming the dimensional structure of the COVID-19-SES and we advocate for SEM to be conducted as a CFA. Consequently, we have included such suggestion in our conclusions. We believe that future studies should build on our findings and test the COVID-19-SES’ dimensionality by performing a CFA after translating the tool to other languages and administering it to samples with different characteristics. In this regard, we have included a new limitation (please see lines 342-344 in the main file) and updated our conclusions (see line 359).

We appreciate that there are many different ways to conduct the psychometric evaluation of a given tool. Different authors and experts in the field of psychometrics use different approaches. In this regard, we followed the guidelines of several highly recognised experts in the field of tool development and psychometric assessment: Streiner, Polit & Beck, Coaley, Norman (please see the reference section in the main file). For these highly cited authors, an instrument’s validity can be tested by means of exploring its content, criterion or construct validity. In order to perform a rigorous assessment of the COVID-19-SES’ validity, we conducted a thorough analysis of these three dimensions of its validity following these experts’ recommendations and methods (please, see references in the main file for each of the methods). In terms of reliability, some of these authors recommend testing the tool’s internal consistency and temporal stability. We not only tested both the COVID-19-SES’ internal consistency (for which we use three different estimators) and temporal stability (test-retest with a 4-week interval), but we also studied the internal consistency of all the COVID-19-SES’ subscales, their correlation between one another and presented the main descriptive data obtained from our data collection as well (as supportive data). We firmly believe that we conducted a methodologically strong study from a data analysis point of view.

Having said all of the above, we cannot perform a CFA as part of our study for the following reasons:

  • CFA is a type of structural equation modelling (SEM) that aims to test a previously established hypothesis (Brown, 2015). CFA is used to test an existing theory. It hypothesizes an a priori model of the underlying structure of the target construct and examines if this model fits the data adequately (Matsunaga, 2010; Polit & Beck, 2020). As the COVID-19-SES is a newly-developed tool, the nature of our study is exploratory and it helps to develop hypothesis that can be tested in future studies (for example: the one related to the three dimensions of the tool).
  • SEM relies on previous knowledge to build the model that is going to be tested (Lee & Song, 2010). This is precisely what this study brings onto the table; our analysis has generated a hypothesis about the COVID-19-SES’ dimensionality that can be tested in future studies.
  • Ideally, variables used in a CFA will have been used in several earlier EFA studies so the psychometric properties and likely factor parameters for the tests are known (Wilaman, 2012).
  • Exploratory Factor Analysis (EFA) is used to generate hypothesis about the dimensionality of a given tool (Raykov & Marcoulides, 2011). EFA is used when researchers have little ideas about the underlying mechanisms of the target phenomena, and therefore, are unsure of how variables would operate vis-à-vis one another (Matsunaga, 2010).. EFA is a tool intended to help generate a new theory by exploring latent factors that best accounts for the variations and interrelationships of the manifest variables (Matsunaga, 2010). This is exactly the data analysis method that we used to assess the COVID-19-SES’ construct validity.
  • CFA is used to test the hypothesis that arise from the EFA (Brown, 2015; Polit & Beck, 2020; Raykov & Marcoulides, 2011; Streiner & Kottner, 2014). Although it could be thought that we should conduct a CFA after having conducted an EFA, this would be a methodological error. CFA must be conducted on a different data set from which the EFA was conducted so that the samples are independent from each other. Failing to do so will lead to p-values that are not trustworthy because capitalization on chance may have occurred (Brown, 2015; Raykov & Marcoulides, 2011. Doing so would be like testing a model on exactly the same sample in which the model was developed.

References

Brown, T. A. (2015). Methodology in the social sciences. Confirmatory factor analysis for applied research (2nd ed.). The Guilford Press. New York.

Lee SY, Song XY. (2010). Structural equation models. International Encyclopedia of Education (3rd ed.). Pages 453-458. doi: https://doi.org/10.1016/B978-0-08-044894-7.01370-1

Matsunaga, M., (2010). How to Factor-Analyze Your Data Right: Do’s, Don’ts, and How-To’s. International Journal of Psychological Research, 3 (1), 97-110.

Polit, D.; Beck, C.T. Nursing Research: generating and assessing evidence for nursing practice, 11th ed.; Wolters Kluwer: Philadelphia, US, 2020.

Raykov, T., & Marcoulides, G. A. (2011). Introduction to psychometric theory. Routledge/Taylor & Francis Group.

Streiner, D.L.; Kottner, J. Recommendations for reporting the results of studies of instrument and scale development and testing. J Adv Nurs 2014, 70, 1970–1979. https://doi.org/10.1111/jan.12402

Widaman, K. F. (2012). Exploratory factor analysis and confirmatory factor analysis. In H. Cooper, P. M. Camic, D. L. Long, A. T. Panter, D. Rindskopf, & K. J. Sher (Eds.), APA handbooks in psychology®. APA handbook of research methods in psychology, Vol. 3. Data analysis and research publication (p. 361–389). American Psychological Association. https://doi.org/10.1037/13621-018

Reviewer’s comment: Second, although the discussion of reliability and validity is discussed, he interpretation of the answer to the question itself is overlooked. It needs more description of the survey results.

Authors’ response: Thanks for your comment. We appreciate that our discussion deepens into the analysis of the COVID-19-SES’s reliability, validity and legibility. We do so because the aim of this study is to develop and psychometrically test the COVID-19 prevention, recognition and home-management self-efficacy scale (COVID-19-SES). This means that this study does not aim to describe and analyse people’s self-efficacy to prevent, recognise and manage COVID-19. The only reason why we include a summary of the participant’s descriptive data is to support the tool’s reliability and validity. We cannot describe the survey results more in depth because this will be the aim of another study we are currently writing for publication elsewhere.

Reviewer 2 Report

Thank you for asking me to review this manuscript.  I am amazed that such study can be completed within such short period of timeline.  Minor comments for considerations:

  • Section 1 (p.2):  ‘protection motivation theory’ and ‘Bandura’s theory’ were included. More information/justifications on how the theory/theories guided the development of the scale is/are suggested.
  • Section 2.4:
    • How was informed consent been obtained from individual participant? Was it done online?
    • more study-specific information on data management and storage for completing ‘the online questionnaire’.  
  • Section 2.6: Was a ‘back-translation’ undertaken? Please provide justifications

Author Response

REVIEWER 2

Reviewer comment: Section 1 (p.2):  ‘protection motivation theory’ and ‘Bandura’s theory’ were included. More information/justifications on how the theory/theories guided the development of the scale is/are suggested.

Authors’ response: Thanks for your comment. Following your recommendation we have included more information on how the ‘protection motivation theory’ and ‘Bandura’s self-efficacy theory’ guided the development of the scale. In the section where we describe how we generated the scale’s items (2.5.1), we have included the following (see the parts highlighted in red in page 3, lines 124-127 and 132-134 of the main file):

According to the PMT, both the effectiveness of recommended behaviours and one’s self-efficacy to perform such behaviours strongly influence people’s motivation to engage in health-preventive conducts [23,24]. In line with the PMT, we generated the COVID-19-SES’ items based on the WHO’s recommended behaviours to protect oneself and others from the spread of COVID-19 [11,20]. Firstly, the research team analysed all the WHO recommendations and identified three categories in which to group the items: (1) prevention of COVID-19 spread and contagion, (2) early recognition of COVID-19 symptoms and (3) home-management of patients with (or suspected) COVID-19 [11,20]. Secondly, we summarised the WHO recommendations and created the initial 19-item version of the COVID-19-SES. The 19 items comprising the COVID-19-SES were created by consensus. Following Bandura’s theory, people’s self-efficacy to adopt protective behaviours was measured in terms of their own capabilities to perform such behaviours [25,26]. In line with Bandura’s self-efficacy theory, the response options ranged from 0 (“completely sure that I cannot do it”) to 100 (“completely sure that I can do it”) [25,36]. In addition, we added some degree of difficulty to all the items in order to avoid ceiling effect (for example: “in any context” at the end of item 4 and “even if my professional or social life are at stake” at the end of item 5) [37].

Reviewer’s comment: Section 2.4: How was informed consent been obtained from individual participant? Was it done online? more study-specific information on data management and storage for completing ‘the online questionnaire’.  

Authors’ response: Thanks for your comment. Following your questions we realised that we needed to clarify this section. Consequently, we have included detailed information about how we gathered the participants’ informed consent and how we managed the data to guarantee their anonymity and confidentiality. Please see lines 108, 111-121 in page 3 of the main file. We included the following:

The study was approved by the competent Ethics and Research Committee (protocol reference code: 76/2020). In the introductory section of the online survey, we provided information about the study (i.e. justification, aim and methods), our research group and participants’ right to withdraw at any point. We also explained how we were going to safeguard their anonymity and confidentiality. All data were treated according to the European legislation on data protection. Only three members of the research team had access to the raw data generated by the online survey platform (M.D.R.-F., I.D.-S. and M.C-C). They were in charge of coupling the participants’ test-retest responses and deleting the participants emails from the initial database. Then, the complete initial database (without email addresses) was handed to two different researchers (I.M.F.-M, M.M.L-R.), who transferred all the data into a SPSS database and randomised the order of the responses. Once the SPSS database was created, the participants’ responses were deleted from the cloud and all the researchers’ laptops. Only the principal researcher has access to the blind SPSS database used for data analysis. All participants accepted voluntary participation before completing the online questionnaire by ticking a box that stated “I agree to participate in the study described above and give my consent for my responses to be used with research purposes”.

Reviewer’s comment: Section 2.6: Was a ‘back-translation’ undertaken? Please provide justifications.

Authors response: Thanks for your comment. We have included detailed information about the translation procedure we followed in order to translate the original COVID-19-SES (in Spanish) into the English version included in the paper. Please see lines 168-176 in page 4 of the main file. We have included the following:

Two independent native English translators (proficient in Spanish) translated the COVID-19-SES from Spanish to English. The small differences between their translations were reconciled by consensus and a single English version of the COVID-19-SES was created. Two independent native Spanish translators (proficient in English) performed a backtranslation of the COVID-19-SES English version into Spanish. Again, small differences in their backtranslation were reconciled by consensus. The research team and the translators examined the original COVID-19-SES, its English translation and the backtranslation in Spanish. It was unanimously considered that the English version of the COVID-19-SES included in this paper respected the semantics of the original tool.

Round 2

Reviewer 2 Report

Thank you for addressing my comments made previously.

Concerns about the terminologies of 'Bandura's self-efficacy theory'. Self-efficacy is a construct of Bandura's Social Cognitive Theory.  Please check and make revisions accordingly.

Further concerns are raised on the revised file - last sentence of  Discussion section 4. If reliability and validity have been tested in this manuscript, why future tests are needed? 

Author Response

Dear reviewer,

Thanks for taking the time to review the revised version of our manuscript. We believe that your comments/suggestions have helped us to improve the overall quality of our paper.

This is our response to the comments you raised after assessing the revised version of the manuscript:

Reviewer's comment: Concerns about the terminologies of 'Bandura's self-efficacy theory'. Self-efficacy is a construct of Bandura's Social Cognitive Theory.  Please check and make revisions accordingly.

Authors' response: Thanks for highlighting this. You are absolute right. We have deleted "Bandura's self-efficacy theory" and added "Bandura's Social Cognitive Theory" (please see lines 69 and 133). Also, we change the sentence "Following Bandura's self-efficacy theory...." for "Following Bandura's recommendations...." (see line 134) because realistically the scoring options are a recommendation by Bandura for authors to develop tools to measure self-efficacy.

Reviewer's comment: Further concerns are raised on the revised file - last sentence of  Discussion section 4. If reliability and validity have been tested in this manuscript, why future tests are needed?

Authors' response: We included this limitation to satisfy a request from another reviewer. However, after reading our limitations section, we realised that the second limitation we included in the manuscript already addresses this issue. What we meant is that further research should test the tool's psychometric properties in order to confirm its reliability and validity in samples with different characteristics. As this is already highlighted in the second limitation of the study, we decided to delete the last sentence of the section 2.4.

We hope this responds to your concerns.

Many thanks,

The Authors